# Description of the Condensed Phases of Water in Terms of Quantum Condensates

**DOI:** 10.3390/e27080885

**Published:** 2025-08-21

**Authors:** François Fillaux

**Affiliations:** MONARIS, CNRS, Campus Pierre et Marie Curie, Sorbonne Université, 4 Place Jussieu, F-75005 Paris, France; francois.fillaux@sorbonne-unversite.fr

**Keywords:** water, hydrogen bonding, quantum condensates, quantum phase transitions, degeneracy entropy

## Abstract

The “abnormal” properties of ice and liquid water can be explained by a hybrid quantum/classical framework based on objective facts. Internal decoherence due to the low dissociation energy of the H-bond and the strong electric dipole moment lead to a quantum condensate of O atoms dressed with classical oscillators and a degenerate electric field. These classical oscillators are either subject to equipartition in the liquid or enslaved to the field interference in the ice. A set of four observables and the degeneracy entropy explain the heat capacities, temperatures, and latent heats of the quantum phase transition; the super-thermal-insulator state of the ice; the transition between high- and low-density liquids by supercooling; AND the temperature of the liquid’s maximum density. The condensate also describes an aerosol of water droplets. In conclusion, quantum condensates turn out to be an essential part of our everyday environment.

## 1. Introduction

Water, the matrix of life, covers two-thirds of our planet and is one of the most abundant substances in the universe. It is of paramount importance in physics and chemistry, earth and life sciences, cosmology, and technology. No other molecule exists as a solid, liquid, or gas at normal pressure. At the microscopic level, the most popular descriptions deal with the classical statistics of distinguishable H_2_O molecules subject to local forces and nuclear quantum effects [1,2,3,4,5,6,7,8,9,10]. Schematically, ice Ih is a frustrated hexagonal lattice of O atoms containing an exponential number of proton configurations according to the “ice rules” [11,12,13], liquid water is a tetra-coordinated network of cooperative H-bonds in a jumble of molecular clusters constantly breaking and forming, and vapor is composed of dimers linked by transient H-bonds.

However, these descriptions do not explain why water is in a class by itself with extraordinary anomalous properties quite different from those found in other materials. Divergent models abound [1,14,15,16], but the physical reality underlying the properties of water is still a mystery, hindering progress in many disciplines.

The aim of this work is to propose a new line of reasoning from microscopic to macroscopic physics, which is convincing in every respect and based solely on objective facts without arbitrary hypotheses. One of the goals is to explain the anomalous evolution of the heat capacity throughout the phase diagram (Figure 1 and Table 1). According to Debye’s model and the energy equipartition theorem, the heat capacity should be proportional to 9R and increase continuously from zero as T→0 to the classical limit 9R as T→∞. Instead, we see that $C_{W}\approx 9R$ is a constant for the liquid in a temperature range far from the infinite limit. It results that the heat capacity is not determined by the thermal density of phonon states. Similarly, $C_{I}=0$ for ice below $T_{0}\approx 8$ K violates Debye’s $T^{3}$ law [17]. Furthermore, the heat capacity of the liquid is halved at $T_{F}$ (fusion) and $T_{B}$ (boiling), violating the equipartition theorem. Another anomaly (not shown) is the dramatic increase in heat capacity by supercooling the liquid to the temperature of homogeneous crystallization at $T_{H}\approx 226$ K [3,18]. Therefore, the description of the heat capacity is beyond the limits of existing statistical models based on equipartition and phonon-mediated heat transfer.

The alternative model presented below is based on the boson character of H_2_O. In condensed matter physics, a Bose–Einstein condensate (BEC) is typically formed when a gas of non-interacting monoatomic bosons at very low density is close to absolute zero. A macroscopic fraction of the atoms described by the same one-particle wavefunction occupy the lowest state, and microscopic quantum phenomena, in particular, wavefunction interference, become macroscopically apparent. Similarly, a quantum condensate of water molecules could correspond to the macroscopic occupation of thermally accessible states. However, there is no theoretical proof of the existence of condensates with complex interactions and multiple internal degrees of freedom. Below, we deduce the existence of water condensates based on physical arguments concerning two H-bond properties: dissociation energy and electric dipole. This inference is validated by its success in explaining the extraordinary properties in question.

This work is structured as follows. The existence of water condensates is justified in Section 2. The dipole eigenstates are introduced in Section 3, where it is shown how quantum entanglement cancels the equipartition. In Section 4, the thermal properties are related to microscopic observables, and the degeneracy entropy is introduced. The gas phase and the aerosol of droplets are treated in Section 5.

## 2. Quantum Condensates

The model consists of a macroscopic number, *N*, of molecules at normal pressure in a sealed container in diathermal equilibrium with a black body at *T*. The density is phase dependent. Boundary effects are negligible. The spin states are degenerate.

At the microscopic level, the existence of stationary states of the H-bonded molecules is precluded by the internal decoherence due to the low dissociation energy of dimers ${\left(H_{2}O\right)}_{2}$ that is typically $D_{0}=\left(1105\pm 10\right)$ cm^−1^ in molecular jets [23]. The calculated potential energy surface shows that $D_{0}$ essentially corresponds to doubly H-bonded dimers [24], so that the dissociation energy of a single H-bond is likely to be about $D_{0}/2$. Since the H-bond dynamics induced by spectroscopy measurements are composed of proton modes above 1600 cm^−1^, librations in the range 400–700 cm^−1^, and O⋯O translations below 200 cm^−1^, $D_{0}/2$ means that these modes are not stationary, as shown by their extremely broad infrared absorption bands [3]. In the absence of quantum measurement, internal decoherence inevitably cancels out nuclear quantum effects and leads to classical proton oscillators.

O atoms dressed with classical protons establish a link to monoatomic condensates. At the macroscopic level, the condensed phases of water are quantum condensates with an electric field due to the strong H-bond dipole (e.g., |μ|≈3 D in the liquid). The many-body wavefunction of the dressed O, $\Phi (r_{1}r_{2}\cdots r_{N},t)$, is symmetric with respect to the exchange $r_{i}⇄r_{j}$ of any two coordinates. This results in a hexagonal structure consisting of honeycomb sheets with a hexagonal unit cell and an atom at its center.

A quantum condensate differs from a monoatomic BEC, such as liquid ^4^He, in that its existence is independent of the thermal wavelength. The number of molecules in the condensate, *N*, is independent of the external temperature, *T*, which is not an internal variable.

## 3. The Dipole States

The classical description of a H-bonded dimer ${\mathrm{HO}}_{d}-H\ldots O_{a}H_{2}$, composed of a donor $H_{2}O_{d}$ and an acceptor $H_{2}O_{a}$, consists of a dimensionless proton in an asymmetric double well along the O⋯O coordinate. The O⋯O length is ≈2.6 Å, and the inter-well separation is ≈0.6 Å. The asymmetry is the energy difference between the ${\mathrm{HO}}_{d}-H\ldots O_{a}H_{2}$ (L) and ${\mathrm{HO}}_{d}\ldots H-O_{a}H_{2}$ (R) configurations, which have opposite dipole moment orientations. Inter-well proton transfer and the dipole flip occur simultaneously.

In their inelastic neutron scattering (INS) studies of ice, Bove et al. [7] fitted the spectra at different temperatures with quasi-elastic profiles and deduced the relaxation rates of thermally activated over-barrier proton jumps. However, they found that the absence of a temperature effect was inconsistent with the model. They concluded that quantum effects were likely but did not pursue this line of research.

Unlike protons, the electric dipole is quantum. It can be represented as a combination of the zero-order electronic states with opposite dipole orientations: $|\mu_{L}\rangle$ and $|\mu_{R}\rangle$, respectively. The dipole states of the L configuration, for example, are(1)$$\begin{array}{ccc} |\mu_{L0}\rangle & = & \cos\phi |\mu_{L}\rangle +\sin\phi |\mu_{R}\rangle ;\;E_{0};\\ |\mu_{L1}\rangle & = & \sin\phi |\mu_{L}\rangle -\cos\phi |\mu_{R}\rangle ;\;E_{0}+\hbar \omega_{1}. \end{array}$$$\hbar \omega_{1}$ is the flipping energy corresponding to the potential asymmetry, and ϕ≪π is the mixing angle. $|\mu_{L0}+\mu_{L1}{|}^{2}$ describes the dipole oscillation at $\omega_{1}$. The amplitude, which is proportional to $2|\mu |\sin^{2}\phi$, is too small to significantly affect the proton residual charges. Thus, energy equipartition applies.

The dipole states of the R configuration, $|\mu_{R0}\rangle$ and $|\mu_{R1}\rangle$, are obtained by swapping $|\mu_{L}\rangle$ and $|\mu_{R}\rangle$. Superposition occurs when the L and R configurations are indistinguishable (e.g., in the gas):(2)$$\begin{array}{ccc} |\mu_{0\pm }\rangle & = & \frac{1}{\sqrt{2}}[|\mu_{L0}\rangle \pm |\mu_{R0}\rangle ];\\ E_{0\pm } & = & E_{0}+\frac{1}{2}\hbar \left(-\omega_{\mu }\pm \omega_{t}\right);\\ |\mu_{1\pm }\rangle & = & \frac{1}{\sqrt{2}}[|\mu_{L1}\rangle \pm |\mu_{R1}\rangle ];\\ E_{1\pm } & = & E_{0}+\hbar \left[\omega_{1}+\frac{1}{2}\left(\omega_{\mu }\pm \omega_{t}\right)\right]. \end{array}$$$\hbar \omega_{\mu }$ is the energy difference between parallel and antiparallel dipoles. $\omega_{t}\approx 2\phi \omega_{1}\ll \omega_{\mu }$ is the beat frequency out of resonance with the normal modes. $|\mu_{0+}+\mu_{0-}{|}^{2}$ and $|\mu_{1+}+\mu_{1-}{|}^{2}$ describe the electronic oscillations with negligible mass and kinetic energy. The huge amplitude proportional to 2|μ| enslaves the classical protons and there is no equipartition. In addition, the O⋯O bond length is stretched by the greater asymmetry compared to (Equation 1), and the H-bond is weakened.

Therefore, Figure 1 is in favor of entangled dipoles (Equation 2) in ice and vapor, and untangled dipoles (Equation 1) in liquid. These findings are in line with neutron scattering measurements.

First, neutron Compton scattering (NCS) probes the mean kinetic energy of the protons. The temperature law expected for equipartition is $\bar{E}\left(T\right)={\bar{E}}_{0}+\frac{3}{2}k_{B}T$. The zero point energy ${\bar{E}}_{0}$ is practically *T* independent, $k_{B}$ is the Boltzmann constant, and $\frac{3}{2}k_{B}\approx 0.12$ meV.mol^−1^.K^−1^. The observed temperature law is quite different [9]. For $T\leq T_{F}$, $\bar{E}=\left(153\pm 2\right)$ meV.mol^−1^ is practically a constant, compared to the expected variation of ≈33 meV.mol^−1^. There is no equipartition (Equation 2). For $T_{F}\leq T\leq T_{B}$, $\bar{E}\left(T\right)\approx {\bar{E}}_{0}+\frac{3}{2}k_{B}\left(T-T_{F}\right)$ means equipartition in the liquid (Equation 1) and freezing of the kinetic energy at the crystallization point $T_{F}$.

Second, for INS, the question is whether the spectra consist of the broad quasi-elastic profile preferred by Bove et al. [7] or, alternatively, whether they consist of INS-induced tunneling transitions at ±(0.10±0.01) meV partially resolved from the elastic peak. The spectra are, prima facie, ambiguous. However, the absence of a temperature effect [7], the split probability density of the protons [25], the NCS data [9], and the heat capacity (see Section 4) are in favor of the tunneling splitting $\hbar \omega_{t}/k_{B}=\left(1.2\pm 0.2\right)\;K$, with semi-subjective error bars.

## 4. The Condensed Phases of Water

### 4.1. Ice Ih

The empirical relation $k_{B}T_{0}\approx 7\hbar \omega_{t}$ (Table 1 and Table 2) gives the tunneling gap of the honeycomb unit cell of the field. Unlike INS, this gap is independent of the measurement, and it does not involve protons. The eigenenergies are seven times those given in (Equation 2), and $\hbar \left(\omega_{1}+\omega_{\mu }\right)/k_{B}$ is proportional to $T_{F}$ (Table 2). The field degeneracy of $\frac{3}{2}$ due to the geometrical frustration calculated by Pauling for an empty hexagon [12] is squared by the atom in the center, which is part of another ring. The ice is a mixture of $\Omega_{I}={\left(\frac{3}{2}\right)}^{2}$ degenerate fields. The entropy $S_{I}=R\ln\Omega_{I}$ is deterministic and independent of temperature.

The tunneling gap is forbidden and $C_{I}\equiv 0$ for $T\leq T_{0}$. Ice can be called a “super-thermal-insulator.” The lowest accessible state by cooling is $RT_{0}$.

For $T_{0}\leq T\leq T_{F}$, the field wavefunction in the occupation-number basis deduced from (Equation 2) is(3)$$\begin{array}{ccc} \Psi_{I}\left(t\right) & = & \sqrt{N_{0-}}\psi_{0-}e^{i\omega_{tI}t}\\ \; & + & \sqrt{N_{1}}\left(\psi_{1+}e^{i\omega_{1+I}t}+\psi_{1-}e^{i\omega_{1-I}t}\right);\\ \frac{N_{0-}}{N_{7}} & = & 1-\Theta_{I};\\ \frac{2N_{1}}{N_{7}} & = & \Theta_{I}; \end{array}$$
where $N_{7}=N/7$; $N_{0-}$ and $N_{1}$ are the occupation numbers; $\omega_{tI}=7\omega_{t}$; $\omega_{1+I}=7\left(\omega_{1}+\omega_{\mu }\right)$; $\omega_{1-I}=7\left(\omega_{1}+\omega_{\mu }+\omega_{t}\right)$. $\Theta_{I}$ is the partition coefficient (Table 3). Apart from its normalization, $\Psi_{I}\left(t\right)$ is the Schrödinger wavefunction, which can be considered a classical quantity without thermal and quantum fluctuations [26]. The probability density $|\Psi_{I}{\left(t\right)|}^{2}$ describes quantum beats corresponding to coherent oscillations of the electric field at $\omega_{BI1}=\omega_{1-I}-\omega_{1+I}$ and $\omega_{BI2}=\omega_{1+I}-\omega_{tI}$. The coherent heat transfer by photons $\hbar \omega_{BI1}$ to the enslaved oscillators with constant kinetic energy gives the heat capacity $C_{I}=\frac{9}{2}R\Theta_{I}$, which is proportional to $T-T_{0}$ (Table 1 and Table 3). According to Plank’s law, the relative power radiated at $\omega_{BI2}$ is ${\left(\omega_{BI2}/\omega_{BI1}\right)}^{3}∼10^{-9}$. This is insignificant.

### 4.2. Liquid Water

The fusion to the high density liquid (HDL) at $T_{F}$ separates each of the $\Omega_{I}$ fields composed of seven entangled dipoles (Equation 2) into 14 degenerate fields of untangled dipoles (Equation 1). The HDL consists of two by two complementary clockwise and counterclockwise honeycomb units composed of XL and (7−X)R, or XR and (7−X)L (X = 1⋯7) configurations. The degeneracy $\Omega_{HD}=14\Omega_{I}$ gives the heat of fusion:(4)$$\Delta F\left(T_{F}\right)=RT_{F}\ln\frac{\Omega_{HD}}{\Omega_{I}}\approx 5993\;J.{\mathrm{mol}}^{-1}.$$This is in reasonable agreement with the measured value of ≈6005 J.mol^−1^ [3,22]. There is no H-bond dissociation or disorder.

Fusion and crystallization are quantum phase transitions. $RT_{F}$ is the field ground state of the liquid. $T_{F}$ is determined by the partition coefficient, which is either $\Theta_{I}=1$ or $\Theta_{HD}=0$. The latent heat is determined by the deterministic degeneracy entropy. Preservation of $\Omega_{I}$ means preservation of the honeycomb structure [27,28]. The HDL is a crystalline liquid, but superfluidity is prevented by the electric field. Internal energy conservation means that the kinetic energy of the liquid ($\frac{9}{2}RT_{F}$) is frozen in the ice crystallized at $T_{F}$.

The eigenstates at $RT_{F}$ and $RT_{B}$ have identical structures with slightly different O⋯O distances. $\hbar \omega_{1}/k_{B}$ is proportional to $T_{F}-T_{B}$ (Table 2). The heat transfer to the classical modes is temperature independent and $C_{HDL}=9R$ (Table 3). By heating, each L → R flip is accompanied by a R → L flip in the complementary unit cell, so the density decreases quadratically. The kinetic energy at the macroscopic level $\frac{9}{2}R\left(T-T_{F}\right)$ is in agreement with NCS at the microscopic level.

The density of liquid water has two puzzling properties: (i) It is maximal at $T_{MD}\approx 4$ K above $T_{F}$ and (ii) it decreases with supercooling. These properties can be explained as follows.

$T_{MD}$ is consistent with the existence of a constant fraction $X_{HD}={\left[12{\left(\frac{3}{2}\right)}^{2}-1\right]}^{-1}\approx 0.038$ of non-interacting complementary hexagons with antiparallel dipole configurations. These dipoles are shielded against thermal waves. By cooling from $T_{B}$, the density reaches a maximum value for $\Theta_{HD}=X_{HD}$, at $T_{MD}=T_{F}\left[1+X_{HD}\left(T_{B}-T_{F}\right)\right]\approx 277$ K, when the occupancy of $RT_{B}$ is completely shielded. $RT_{MD}$ is the lowest HDL field state accessible by cooling, and $T_{MD}$ is the critical temperature for the onset of the supercooled liquid.

Below $T_{MD}$, the energy gain ($\hbar \omega_{\mu }$) of the antiparallel dipoles of the complementary units favors non-degenerate units with zero dipole moment ($\Omega_{LD}=1$), which cluster into the ground state of the low-density liquid (LDL) field at $RT_{H}=R\left(T_{F}-\hbar \omega_{\mu }/k_{B}\right)$ (Table 2). $T_{H}\approx 226$ K is the temperature of homogeneous crystallization reported in various papers in the range of 226−232 K [3,18]. The supercooled liquid is a mixture of $\Theta_{SC}$HDL and $\left(1-\Theta_{SC}\right)$LDL (Table 3). Supercooling is a continuous quantum transition between the ground states of the HDL and LDL fields whose internal energy gain cannot be radiated away (Table 3). The heat capacity $C_{SC}=R\left(9+\Theta_{SC}^{2}\ln\Omega_{HD}\right)$ increases from 9R≈75 J.mol.^−1^$T^{-1}$ at $T_{MD}$ to about 12.5R≈103 J.mol.^−1^$T^{-1}$ at $T_{H}$, in agreement with the measurements [18], and the density decreases quadratically. Homogeneous crystallization at $T_{H}$ is a quantum transition from the ground state of the LDL untangled field, whose latent heat is removed from the liquid. The frozen kinetic energy of ice crystallized at $T_{H}$ is $\frac{9}{2}RT_{H}$.

As a result, the condensed phases are isomorphic. They differ in that the kinetic energy in the liquid allows collective excitations that are forbidden for enslaved oscillators in the solid. In contrast, the degeneracy is not critical for the physical state.

## 5. Other Phases of Water

### 5.1. Gaseous Water

Ebullition at $T_{B}$ and evaporation below $T_{B}$ can be treated on the same foot. The molar volume expansion of ≈1700 at normal pressure dissociates six of the seven H-bonds per unit cell and destroys the condensate. This leads to a gas of distant entangled dimers (Equation 2) whose tunneling splitting is $\hbar \omega_{t}/k_{B}\approx 0.94$ K [29]. The heat capacity $\frac{9}{2}R$ (Table 1) means that $\hbar \omega_{t}$ is independent of the act of measurement.

The energy of the transition is(5)$$\begin{array}{ccc} \Delta F_{G}\left(T\right) & = & 6D_{0}-RT\left(9+\ln\Omega_{HD}\right). \end{array}$$The heat of ebullition $\Delta F_{G}\left(T_{B}\right)=\left(40,575\pm 700\right)\;J.{\mathrm{mol}}^{-1}$ agrees with the measured value of $\left(40,660\pm 80\right)\;J.{\mathrm{mol}}^{-1}$ [30], and the measured heat of sublimation of ice at $T_{F}$, namely, ≈51,059 J.mol^−1^ [21], matches $\Delta F_{F}+\Delta F_{G}\left(T_{F}\right)\approx 51,058$ J.mol^−1^ for a two-step process through the liquid state.

There are two points worth noting. First, the $\Omega_{I}$ value for the honeycomb structure differs from Pauling’s estimate for the empty hexagonal structure. Second, the dissociation energy of the untangled H-bonds in the liquid is exactly $D_{0}/2$. Thus, the entanglement (Equation 2) is energy free. This means that the electric energy gain $\hbar \omega_{\mu }$ is compensated by the weakening of the O⋯O bond. The H-bond is essentially electrostatic in nature, with no directional valence bond energy [24], and there is no significant cooperative effect in the condensed phases.

### 5.2. Water Droplets

The condensation of vapor over the locally heated water surface can produce long-lived, self-organizing aggregates with a honeycomb pattern on the millimeter scale. These aggregates are composed of equidistant monodisperse droplets with radii ranging from 5 to 100 μm. This single layer floats above the surface, and the spatial order is maintained even as the layer moves horizontally [31,32,33,34,35]. The self-assembly mechanism is thought to be a balance of hydrodynamic forces. However, while the upward flow of steam can explain the repulsive force, there is no convincing explanation for the attractive forces between droplets.

In fact, it is unlikely that attractive and repulsive forces of different natures will cancel each other out. The absence of attractive forces is necessary for the formation of long-lived droplets, while the absence of repulsive forces is necessary for the formatio of their long-lived structure. The honeycomb pattern is consistent with a BEC of non-interacting bosons. When droplets emerge from vapor with zero center-of-mass kinetic energy, extended de Broglie wave interference yields a hexagonal pattern in 3D. However, the droplets experience two vertical forces without horizontal components, which destroys the vertical interference and yields the honeycomb layer. The downward force due to gravity is proportional to $r^{3}$, and the upward force due to the vapor flow is proportional to $r^{2}$. At a critical radius, $r_{c}\left(T\right)$, the honeycomb layer composed of monodisperse droplets moves downward toward the free surface. This two-dimensional supersolid can flow as a superfluid with zero viscosity. In the reported experiments, the layer hinders upward vapor flow from the free surface and prevents the formation of multiple layers. Nevertheless, large-scale, three-dimensional condensates can account for long-lived clouds drifting in the troposphere.

## 6. Conclusions

The physical reality underlying the H-bonded dimer in the absence of quantum measurement is the coexistence of the classical degrees of freedom of the water molecules and the quantum states of the electric dipole. The condensed phases are condensates of O atoms dressed with classical oscillators in a degenerate electric field. The classical oscillators are either subject to equipartition in the liquid or enslaved to the electric field with negligible kinetic energy in the ice.

This description is based on objective facts, without arbitrary hypotheses, and it is parsimonious. It captures properties that lie outside the bounds of classical statistics and thermodynamics by a set of four microscopic observables and degeneracy entropy. It explains why water is in a class by itself with extraordinary properties. It shows that the hexagonal structure is not due to local forces, and that the difference between the liquid and solid states is essentially a matter of kinetic energy, not of disorder.

Compared to monoatomic systems, the condensed phases of water and the aerosol extend the size, temperature, and density ranges of condensates by orders of magnitude to the scale of our everyday environment under standard conditions. This is likely to change our understanding of the chemistry in water and how life works.

## Figures and Tables

**Figure 1 entropy-27-00885-f001:**
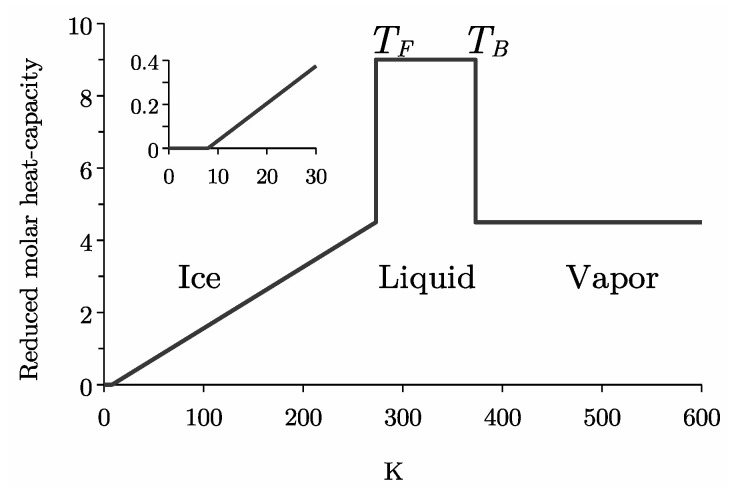
Reduced molar heat-capacity C/R of H_2_O (Table 1).

**Table 1 entropy-27-00885-t001:** Reduced molar heat capacities of the phases of water. R≈8.314 J.mol^−1^.K^−1^. $T_{0}=\left(8\pm 1\right)$ K. $T_{F}\approx 273.16$ K. $T_{B}\approx 373.16$ K. * Neutron scattering demonstrates ice Ih at 5 K [7,8,9].

		2C/(9R)	Ref.
Vapor	$T_{B}\leq T$	1.001	[19]
Liquid	$T_{F}\leq T\leq T_{B}$	2.02	[20]
Ice Ih	$T_{0}\leq T\leq T_{F}$	$1.01\frac{T-T_{0}}{T_{F}-T_{0}}$	[21,22]
Ice Ih *	$0\leq T\leq T_{0}$	$<10^{-2}$	[17]

**Table 2 entropy-27-00885-t002:** Critical temperatures and microscopic observables. $\hbar \omega_{t}/k_{B}=\left(1.2\pm 0.2\right)$ K; $\hbar \omega_{1}/k_{B}\approx 129$ K; $\hbar \omega_{\mu }/k_{B}\approx 47$ K.

$\hbar \omega_{t}/k_{B}$	$\hbar \omega_{1}/k_{B}$	$\hbar \left(\omega_{1}+\omega_{\mu }\right)/k_{B}$	$\hbar \omega_{\mu }/k_{B}$
$\frac{T_{0}}{7}$	$\frac{9}{7}\left(T_{B}-T_{F}\right)$	$\frac{9}{14}T_{F}$	$T_{F}-T_{H}$

**Table 3 entropy-27-00885-t003:** Energies, E, and partition coefficients, Θ, of the quantum phases of water. $T_{0}\approx 8$ K; $T_{F}\approx 273$ K; $T_{B}\approx 373$ K; $T_{H}\approx 226$ K; $T_{MD}\approx 277$ K. $\Omega_{HD}=14{\left(\frac{3}{2}\right)}^{2}$. HDL: high density liquid. SC: supercooled.

		E/R	Θ
$T_{B}\leq T\leq T_{c}$	Steam	$\frac{9}{2}\left(T_{B}+T\right)$	—
$T_{F}\leq T\leq T_{B}$	HDL	9T	$\Theta_{HD}=\frac{T_{B}-T}{T_{B}-T_{F}}$
$T_{H}\leq T\leq T_{F}$	SC	$(9+\Theta_{SC}^{2}\ln\Omega_{HD})T$	$\Theta_{SC}=\frac{T_{F}-T}{T_{F}-T_{H}}$
$T_{0}\leq T\leq T_{F}$	Ice Ih	$\frac{9}{2}\left(T_{F}+\Theta_{I}T\right)$	$\Theta_{I}=\frac{T-T_{0}}{T_{F}-T_{0}}$

## Data Availability

Data is contained within the article.

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
