# Peer review of "Description of the Condensed Phases of Water in Terms of Quantum Condensates"

_entropy, 2025, doi:10.3390/e27080885_

Round 1

Reviewer 1 Report

Comments and Suggestions for Authors

Based on the manuscript "DESCRIPTION OF THE CONDENSED PHASES OF WATER IN TERMS OF QUANTUM CONDENSATES"

Recommendation: Major Revisions

Summary:
The author presents a provocative and unconventional theory aiming to describe the condensed phases of water as quantum condensates. While intellectually stimulating, the manuscript suffers from fundamental issues in scientific rigor, clarity, and substantiation of its claims.

Major Concerns:

  1. Lack of Empirical Validation:
    The model relies heavily on speculative quantum mechanical constructs (e.g., dipole entanglement, zero kinetic energy in ice, coherent quantum beats in bulk water) without presenting experimental data, quantitative simulations to support the claims or strong and robust derivations. 

  2. Overextension of the Quantum Condensate Concept:
    The manuscript extrapolates the notion of Bose-Einstein condensation (BEC) far beyond the commonly accepted limits. BECs in complex molecular systems at room temperature are not established, and the connection drawn to water lacks a rigorous derivation. This weakens the scientific plausibility.

  3. Unjustified Dismissal of Established Models:
    The paper dismisses existing models of water behavior (based on classical molecular dynamics and nuclear quantum effects) with insufficient critical analysis. It would benefit from a balanced discussion of how the proposed model compares and complements (or contradicts) established theories.

  4. Insufficient Mathematical Formalism:
    Despite its reliance on quantum principles, the manuscript lacks the mathematical rigor necessary to validate or test the model quantitatively. The wavefunctions, for instance, are introduced without derivation or context. Derivations, that are expected, are reduced to the introduction of disconnected equations usually reported in a model-dependent form. Many of the theoretical arguments sounds speculative.

Minor Concerns:

  • Figures and Tables: Figure 1 and Table 1 are discussed with minimal clarity. The graphical representation should be supported with captions that help readers understand the data source, relevance, and interpretation.

  • Grammar & Language: Some sentences are awkwardly structured, and the flow between sections is sometimes disjointed.

Suggestions for Improvement:

  • Include numerical simulations or model predictions that can be tested against empirical data.

  • Reframe speculative ideas with clear disclaimers and identify testable hypotheses.

  • Improve the precision of language and avoid terms that might be perceived as sensational.

  • Contextualise this framework within the broader body of work on quantum water models.

Conclusion:

While the manuscript introduces original and potentially groundbreaking ideas, it lacks the necessary rigor, empirical support, and clarity to be accepted in its current form. I do not suggest publication.

Author Response

1 Lack of Empirical Validation:
The model relies heavily on speculative quantum mechanical constructs (e.g., dipole entanglement, zero kinetic energy in ice, coherent quantum beats in bulk water) without presenting experimental data, quantitative simulations to support the claims or strong and robust derivations. 

Response 1:

This comment does not make any sense at all. It is shocking to read that the model has no contact with experimental data, since the argumentation is based solely on objective experimental facts without arbitrary hypotheses. At the microscopic level, equations (1) and (2) are confronted with neutron scattering data. At the macroscopic level, the ice wavefunction (3) is based on the empirical relation kB T0 = 7ℏωt , and quantum beats quantitatively explain the linear variation of the heat capacity proportional to T − T0 . The hexagonal structure and the calculated latent heats are compared with measurements. The model is purely analytical. It is beyondthe limits of statistical physics based on molecular dynamics simulations.

Action: p. 2 l 42-43: The description of the heat capacity is beyond the limits of existing statistical models based on equipartition and phonon-mediated heat transfer.

Overextension of the Quantum Condensate Concept:
The manuscript extrapolates the notion of Bose-Einstein condensation (BEC) far beyond the commonly accepted limits. BECs in complex molecular systems at room temperature are not established, and the connection drawn to water lacks a rigorous derivation. This weakens the scientific plausibility.

Response:

The existence of quantum condensates is logically deduced from two undeniable facts: the boson character of water molecules and the internal decoherence of the H-bond. The notion of dressed O atoms establishes a direct link to monoatomic BECs. This is strictly logical, without any arbitrary hypotheses.

Action: p 3 line 75: O atoms dressed with classical oscillators establish a link to monoatomic BECs.

Unjustified Dismissal of Established Models:
The paper dismisses existing models of water behavior (based on classical molecular dynamics and nuclear quantum effects) with insufficient critical analysis. It would benefit from a balanced discussion of how the proposed model compares and complements (or contradicts) established theories.

Response: The third paragraph starting at line 30 shows that the heat capacity is definitely in conflict with all existing models based on phonon mediated heat transfer and equipartition, irrespective of their theoretical background.

Action: p 2 lines 42-43: The description of the heat capacity is beyond the limits of existing
statistical models based on equipartition and phonon-mediated heat transfer.

Insufficient Mathematical Formalism:
Despite its reliance on quantum principles, the manuscript lacks the mathematical rigor necessary to validate or test the model quantitatively. The wavefunctions, for instance, are introduced without derivation or context. Derivations, that are expected, are reduced to the introduction of disconnected equations usually reported in a model-dependent form. Many of the theoretical arguments sounds speculative.

Response: The context of this work is clearly the H-bond between water molecule, and the existence of two possible dipole orientations is widely accepted. (See Ref. 3 for further justification.) There is nothing new in (1), and (2) is a logical continuation.

No action.

Minor Concerns:

  • Figures and Tables: Figure 1 and Table 1 are discussed with minimal clarity. The graphical representation should be supported with captions that help readers understand the data source, relevance, and interpretation.

  • Fig. 1 is too simple to be ambiguous. Table 1 gives all definitions and references.
  • No action.
  • Grammar & Language: Some sentences are awkwardly structured, and the flow between sections is sometimes disjointed.
  • No response.

Reviewer 2 Report

Comments and Suggestions for Authors

Review of the article

DESCRIPTION OF THE CONDENSED PHASES OF WATER IN TERMS OF QUANTUM CONDENSATES

François Fillaux

  • Objective of the paper
  • Proposal of a new line of reasoning from microscopic to macroscopic physics
  • Explanation of the anomalous evolution of the heat capacity of water
  • Anomalous heat capacity of water
  • Heat capacity of water does not correspond to the Debye model and the theorem of equipartition of energy
  • Heat capacity is not determined by the thermal density of phonon states
  • Heat capacity of ice below T0 ≈ 8 K violates Debye's law T3
  • Bosonic character of H2O
  • The starting point of the new argument is the bosonic character of H2O
  • Bose-Einstein condensate (BEC) is formed at low density and temperature close to absolute zero
  • Macroscopic fraction of atoms occupy the lowest state, making quantum phenomena macroscopically obvious
  • Quantum condensate and hydrogen bonds
  • Quantum condensate refers to the macroscopic occupation of multiple thermally accessible states
  • Proving the existence of BEC with complex interactions and internal degrees of freedom is a major open problem
  • In the paper, the existence of condensates is deduced from the properties of hydrogen bonds and the electric dipole
  • Hybrid quantum-classical structure
  • Internal decoherence due to low dissociation energy and strong electric dipole moment leads to a quantum condensate of O atoms
  • Classical oscillators are either normal modes in liquid or subject to massless field interference in ice
  • Explanation of anomalous properties of water
  • Heat capacities, temperatures, and latent heats of quantum phase transitions are explained by a set of four observables and degeneracy entropy
  • Condensate also describes an aerosol of water droplets
  • Interest and significance of the article
  • The article deserves attention for the interesting approach of the author
  • It can be published, it is of interest to scientists in this area of ​​science
  • The article does not mention the nuclear quantum effect and its role in the properties of water.

Author Response

This review is a fair representation of my work, which means that my arguments are quite understandable to an open-minded reader.

Regarding the last sentence: The article does not mention the nuclear quantum effect and its role in the properties of water.

Response: Nuclear quantum effects are mentioned in the introduction, but they are canceled out by internal
decoherence.

Action: p. 3, line 73: In the absence of quantum measurement, internal decoherence inevitably cancels out nuclear quantum effects, and leads to classical modes.

Reviewer 3 Report

Comments and Suggestions for Authors

The paper titled "Description of the Condensed Phases of Water in Terms of Quantum Condensates" presents a study to interpret the anomalous thermodynamic behavior of water using concepts from quantum mechanics, such as dipole entanglement, degeneracy entropy, and quantum condensates. While the paper aims to offer a novel unifying picture, it falls short in several key areas related to physical justification, methodological rigor, and clarity. For the manuscript to meet the standards, it requires substantial revision and a firmer grounding in established theory.
The manuscript introduces a hybrid model of O atoms "dressed" with classical oscillators and entangled dipolar fields, suggesting that this structure constitutes a form of quantum condensate across the condensed phases of waterThere is no clearly defined mathematical formalism or quantum statistical framework to support this interpretation. The absence of a Hamiltonian, explicit many-body wavefunctions, or variational arguments significantly limits the credibility of the proposed model.
Several key terms—such as "quantum condensate," "superinsulator," and "massless dipole"—are used in a non-standard manner so this needs further explanation. 
The claims regarding zero kinetic energy in ice and the enslavement of classical oscillators by dipolar fields are conceptually interesting but lack empirical or computational support. The author references heat capacity data and neutron scattering measurements but does not provide simulations, model fitting, or comparisons with alternative models.
The treatment of phase transitions as quantum in nature is provocative but not rigorously developed. The derivations rely on heuristic entropy arguments and partition coefficients rather than standard thermodynamic or quantum statistical mechanics. A clearer distinction should be made between metaphorical and literal uses of quantum concepts.
Equations luck sufficient explanatory context, making them inaccessible to many readers. Clarifying the physical meaning of each parameter and simplifying the narrative flow would improve readability. Similarly, the discussion on self-assembled water droplets and atmospheric condensates introduces interesting phenomena but remains speculative and unsupported by quantitative modeling.
The manuscript would be strengthened by comparing the proposed theory to the existing theories including classical lattice models, classical water model simulations and quantum Monte Carlo studies of hydrogen bonding. 

Author Response

Comment 1: The paper titled "Description of the Condensed Phases of Water in Terms of Quantum Condensates" presents a study to interpret the anomalous thermodynamic behavior of water using concepts from quantum mechanics, such as dipole entanglement, degeneracy entropy, and quantum condensates. While the paper aims to offer a novel unifying picture, it falls short in several key areas related to physical justification, methodological rigor, and clarity. For the manuscript to meet the standards, it requires substantial revision and a firmer grounding in established theory.

Response 1: "physical justification", each numerical value is conforted by empirical data; "methodological rigor", the referee does not point out any logical flaws; "clarity", suggestions would be appreciated.   

Comment 2: The manuscript introduces a hybrid model of O atoms "dressed" with classical oscillators and entangled dipolar fields, suggesting that this structure constitutes a form of quantum condensate across the condensed phases of waterThere is no clearly defined mathematical formalism or quantum statistical framework to support this interpretation. The absence of a Hamiltonian, explicit many-body wavefunctions, or variational arguments significantly limits the credibility of the proposed model.

Response 2: There is no H-bond Hamiltonian. The many-body wavefunction symmetry page 3 line 78, and the field wavefunction (3) are sufficient. The variational argument is the dissociation energy. The credibility of the condensate is that it explains anomalous properties, contrary to existing models. 

Action 2 page 2 line 52 : "which is practically intractable in the case of water for which there is no H-bond Hamiltonian.

Comment 3: Several key terms—such as "quantum condensate," "superinsulator," and "massless dipole"—are used in a non-standard manner so this needs further explanation.

Response 3: "quantum condensate" see p 3 l 82: "A quantum condensate differs from a monoatomic BEC, such as liquid helium, in that its existence is independent of the thermal wavelength. Since the dissociation energy excludes the molecular states of the ordinary phase at any temperature, the quantum condensate contains a fixed number of molecules, N, regardless of the external temperature, T, which is not an internal variable. Thus, contrary to liquid 4 He, there is no statistical fluctuation of the occupation number."; "superinnsulator" see p 5 l 146: "Since the tunneling gap is forbidden, C I ≡ 0 for T ≤ T0 , contradicting Debye’s law. Ice can be called a “superinsulator”. The lowest accessible state by cooling is R T0."; "massless dipole" see p 4 l 115: "of electric charges with negligible masses".

Comment 4: The claims regarding zero kinetic energy in ice and the enslavement of classical oscillators by dipolar fields are conceptually interesting but lack empirical or computational support. The author references heat capacity data and neutron scattering measurements but does not provide simulations, model fitting, or comparisons with alternative models.

Response 4: In the introduction p 1 l 43... I show that Fig. 1 is at variance with Debye's model and energy equipartition. p 4, the INS and NCS data were already analyzed in ref [7] and [9], respectively. I am not aware of any alternative model contradicting equipartition.

Comment 5: The treatment of phase transitions as quantum in nature is provocative but not rigorously developed. The derivations rely on heuristic entropy arguments and partition coefficients rather than standard thermodynamic or quantum statistical mechanics. A clearer distinction should be made between metaphorical and literal uses of quantum concepts.

Response 5: Degeneracy entropy is not "heuristic." Rather, it is logically deduced from the honeycomb structure. Standard thermodynamic and quantum statistical mechanics do not explain the anomalous properties of water. For a quantum condensate, a classical phase transition driven by a competition between energy and entropy of thermal fluctuations is excluded. The phase transition of water is driven by the non-temperature partition coefficient. It is literally quantum, not just metaphorically. 

Comment 6: Equations luck sufficient explanatory context, making them inaccessible to many readers. Clarifying the physical meaning of each parameter and simplifying the narrative flow would improve readability. 

Response 6: Requests for more explanatory context and a simpler narrative seem contradictory. The five equations are straightforward, and I don't see any ambiguity. There is no parameter in (1) and (2). Only observables.  

Comment 6': Similarly, the discussion on self-assembled water droplets and atmospheric condensates introduces interesting phenomena but remains speculative and unsupported by quantitative modeling. 

Response 6': The classical models presented in refs [33-37] fail to explain the stability of self-assembled water droplets. 

Comment 7: The manuscript would be strengthened by comparing the proposed theory to the existing theories including classical lattice models, classical water model simulations and quantum Monte Carlo studies of hydrogen bonding. 

Respose 7: There are numerous theoretical works. However, none of them explain the breakdown of equipartition.

Reviewer 4 Report

Comments and Suggestions for Authors

The presented work continues the author’s research in the field of quantum phase transitions of water. It is shown that some anomalous properties of liquid water and ice can be explained in a combined quantum/classical framework.

The manuscript is clear and presented in a well-structured manner. Presented research relevant for the field. The cited references are mostly recent publications.
But from my point of view this paper have room for improvement.

-Due to the importance of the conclusions, it is necessary to present the dipole states in more details (I did not find this in references [25,26]) and maybe provide this part of the article with supplemented materials.

-It is not clear what is meant in section 5.2. The droplet size in 2D clusters varies within 5-100 micrometers. Cluster sizes vary within even larger limits. It is not clear how quantum properties are extended to the macro object and what interactions are responsible for the attraction of droplets.

Author Response

Comment 1: "Due to the importance of the conclusions, it is necessary to present the dipole states in more details (I did not find this in references [25,26]) and maybe provide this part of the article with supplemented materials."

Response: see Sec. 3.

Comment 2: "It is not clear what is meant in section 5.2. The droplet size in 2D clusters varies within 5-100 micrometers. Cluster sizes vary within even larger limits. It is not clear how quantum properties are extended to the macro object and what interactions are responsible for the attraction of droplets."

Response 2: See Sec. 5.2 

Reviewer 5 Report

Comments and Suggestions for Authors

The manuscript suffers from multiple occurrences of misunderstanding in all aspect the authors tried to consider. Phrases like "In the absence of quantum measurement, internal decoherence 
inevitably cancels out nuclear quantum effects, and leads to classical modes.", "O atoms dressed with classical oscillators establish a link to monoatomic BECs", "A quantum condensate differs from a monoatomic BEC, such as liquid 4He, in that its existence is independent of the thermal wavelength" and many more similar meaningless combinations of words make it impossible to grasp the meaning of this manuscript. This manuscript must be rejected and the author should perhaps spend more time trying to explain what he wanted to say. The quality of text, printout full of crossed words, etc. appears quite disrespectful to the journal and its readership.

Round 2

Reviewer 2 Report

Comments and Suggestions for Authors

Manuscript has been sufficiently improved to warrant publication in Entropy.

Author Response

Manuscript has been sufficiently improved to warrant publication in Entropy.

I hope the editors follow this recommendation.

Reviewer 3 Report

Comments and Suggestions for Authors
  • While the matching of empirical data is appreciated, this alone does not satisfy the need for physical justification. The author dismisses the lack of clarity too readily — the paper introduces novel concepts without sufficient explanatory scaffolding (e.g., figures, glossary, narrative flow). Include a clear conceptual roadmap, define all terms up front, and add summary tables/figures that link empirical data to theoretical components.
  • The absence of a Hamiltonian may be unavoidable due to the complexity of H-bonds in water, but then the author should construct an effective or phenomenological Hamiltonian. Equation (3) is a heuristic occupation-based expression. The dissociation energy is an observable, not a variational parameter in any formal sense. Introduce an effective Hamiltonian or justify its absence more thoroughly by referencing relevant limitations in quantum statistical mechanics of complex fluids.

  • The terms are defined, but their use diverges significantly from standard physics and quantum statistical mechanics. For example, “quantum condensate” here does not require thermal wavelength conditions or occupation fluctuations, which contradicts conventional usage. “Superinsulator” and “massless dipole” are also non-standard and will confuse most readers. Add a glossary or explanatory box distinguishing these definitions from standard usage and clarifying why alternative terms are chosen.

  • The cited data (e.g., Ref [9]) does suggest deviations from classical equipartition. Either moderate claims or provide a supplementary figure comparing experimental data directly with model predictions, including uncertainties.

  • While the use of geometrical entropy is a legitimate construct, the quantum nature of the phase transitions is not demonstrated via formalism. The distinction between “literal” and “metaphorical” uses of “quantum” is still ambiguous. Clarify what is meant by "quantum" in this context — is it merely structural (fixed N, tunneling gaps), or is there a time-evolution quantum coherence being claimed?

  • The response is dismissive. Many variables in Eqs. (1)–(3), such as ωt, ωμ, ΘI, etc., appear without dimensional analysis, derivation, or explicit physical interpretation. Add a table summarizing all variables with definitions, units, physical meaning, and how they are obtained. Include brief derivations or references for key expressions.

  • This section is highly speculative and does not belong in a central argument about condensed matter phases unless it is backed by quantitative modeling or empirical data. Dismissing other models is not a substitute for offering a robust alternative.

  • Simply asserting the superiority of a model is insufficient — quantitative comparison with at least 2–3 mainstream models (e.g., TIP4P, MB-pol, QMC hydrogen-bond studies) is expected in a scientific paper. Add a short comparative table summarizing how major models perform against key anomalies (e.g., heat capacity, density max, entropy) and where this model fits in.

Author Response

Comment 1: "While the matching of empirical data is appreciated, this alone does not satisfy the need for physical justification. The author dismisses the lack of clarity too readily — the paper introduces novel concepts without sufficient explanatory scaffolding (e.g., figures, glossary, narrative flow). Include a clear conceptual roadmap, define all terms up front, and add summary tables/figures that link empirical data to theoretical components."

Response 1: Matching empirical data justifies legitimacy. (i) The concepts of "super-thermal-insulator" for ice, and that of degeneracy entropy are defined p 5 l 148 and l 147, respectively. (ii) There is only one figure whose interpretation is straightforward (Table 1). (iii) The roadmap is presented p 1 l 30-32: "The aim of this work is to propose a new line of reasoning from microscopic to macroscopic physics, which is convincing in every respect and based solely on objective facts without arbitrary hypotheses." The structure of the paper is presented p 2 from l 87 on.(iv) Tables 2 and 3 link the empirical data to observables. 

Action 1: "super-thermal-insulator" replaces "superinsulator".

Comment 2': "The absence of a Hamiltonian may be unavoidable due to the complexity of H-bonds in water, but then the author should construct an effective or phenomenological Hamiltonian. ... Introduce an effective Hamiltonian or justify its absence more thoroughly by referencing relevant limitations in quantum statistical mechanics of complex fluids."

Response 2': This is irrelevant. The H-bond does not have a specific Hamiltonian that can be separated from the molecular Hamiltonian.  

Action 2': p 2 l 49 "In this context, a quantum condensate of water molecules is characterized by the macroscopic occupation of multiple thermally accessible states. Proving the existence of such condensates with complex interactions and internal degrees of freedom is beyond the scope of this paper. Nevertheless, one can infer the existence of condensates based on two H-bond properties: dissociation energy and electric dipole. This inference can be validated by its ability to explain the extraordinary properties in question.

Comment 2''': "Equation (3) is a heuristic occupation-based expression." 

Response 2''': This is controversial. For Leggett [28] it is classical. 

Comment 2''': "The dissociation energy is an observable, not a variational parameter in any formal sense." 

Response 2''': I agree (see conclusion). 

Comment 3: "The terms are defined, but their use diverges significantly from standard physics and quantum statistical mechanics. For example, “quantum condensate” here does not require thermal wavelength conditions or occupation fluctuations, which contradicts conventional usage. “Superinsulator” and “massless dipole” are also non-standard and will confuse most readers. Add a glossary or explanatory box distinguishing these definitions from standard usage and clarifying why alternative terms are chosen. 

Response 3: For quantum condensate see p 3 l 82. Superinsulator is now "super-thermal-insulator" p 5 l 148. For massless dipole see

Action 3: p 4 l 117 |ψ0+ + ψ0− |2 and |ψ1+ + ψ1− |2 describe the quantum beat of electronic charges with negligible masses compared to nuclei, whose frequency ωt is out of resonance with the normal modes. The huge amplitude proportional to 2|µ| enslaves the classical oscillators to the electronic charges, which have negligible kinetic energy. There is no equipartition.

Comment 4: "The cited data (e.g., Ref [9]) does suggest deviations from classical equipartition. Either moderate claims or provide a supplementary figure comparing experimental data directly with model predictions, including uncertainties."

Action 4: p 4 l 127 "For T ≤ TF , Ē = (153 ± 2) meV.mol−1 is practically a constant, compared to the expected variation of
≈ 33 meV.mol−1 . There is no equipartition, in agreement with (2)."

Comment 5: "While the use of geometrical entropy is a legitimate construct, the quantum nature of the phase transitions is not demonstrated via formalism. The distinction between “literal” and “metaphorical” uses of “quantum” is still ambiguous. Clarify what is meant by "quantum" in this context — is it merely structural (fixed N, tunneling gaps), or is there a time-evolution quantum coherence being claimed?"

Action 5 p 6 l 170: "Since R TF is the field ground state of the liquid, the transition is exclusively driven by non-temperature
entropy without H-bond dissociation or disorder. It is purely quantum."

Comment 6: "The response is dismissive. Many variables in Eqs. (1)–(3), such as ωt, ωμ, ΘI, etc., appear without dimensional analysis, derivation, or explicit physical interpretation. Add a table summarizing all variables with definitions, units, physical meaning, and how they are obtained. Include brief derivations or references for key expressions."

Response 6: This is unfair; h̄ω1 is the flipping energy (p 3 l 106); h̄ωµ is the energy difference between parallel and antiparallel dipoles (p 4 l 115); ωt ≈ 2ϕω1 ≪ ωµ is the tunneling frequency (p 4 l 115); Θ is the partition coefficient (p 5 l 154 and (3)); Tables 2 and 3 give all observables and variables values with definitions and units.  

Comment 7: "This section is highly speculative and does not belong in a central argument about condensed matter phases unless it is backed by quantitative modeling or empirical data. Dismissing other models is not a substitute for offering a robust alternative." 

Response 7: The droplet aerosol can still be considered a phase of water, even though condensate is not the ultimate solution. 

Comment 8': "Simply asserting the superiority of a model is insufficient — "

Response 8':  This is unfair. There is no assertion of any superiority. On page 7, line 253, it says, "This description is based on objective facts, without arbitrary hypotheses, and it is much more parsimonious than the other models." This is a fact. However, I agree that "other models" can be omitted.

Action 8': p 7 l 251 read "This description is based on objective facts, without arbitrary hypotheses, and it is parsimonious."

Comment 8'': "quantitative comparison with at least 2–3 mainstream models (e.g., TIP4P, MB-pol, QMC hydrogen-bond studies) is expected in a scientific paper. Add a short comparative table summarizing how major models perform against key anomalies (e.g., heat capacity, density max, entropy) and where this model fits in."

Response 8'': There are more than 4,000 published papers on water. There are over 70 anomalous properties. There are dozens of models. It is impossible to objectively and clearly summarize this information in a short table. Additionally, comparison is insignificant since the conceptual frameworks are completely different.   

Round 3

Reviewer 3 Report

Comments and Suggestions for Authors

While the manuscript introduces a novel interpretative framework for water’s anomalies, the author's responses do not adequately address several core concerns raised in the review. Moreover, the tone of the responses is often dismissive rather than constructively engaging with scientific critique. If the author is unable or unwilling to provide clear, rigorous, and conceptually sound responses to foundational questions, I recommend that the manuscript be rejected.

Author Response

Referee 3 requests an unrealistic H-bond Hamiltonian, suggesting a miscon-
ception of this peculiar chemical bond. Thus, the main reason for rejection
is a conceptual error. Additionally, the instruction Add a short comparative
table summarizing how major models perform against key anomalies does
not make sense. By definition, anomalies are not explained by the existing
models.